# Unifying Bilateral Filtering and Adversarial Training for Robust Neural Networks

## Abstract

Recent analysis of deep neural networks has revealed their vulnerability to carefully structured adversarial examples. Many effective algorithms exist to craft these adversarial examples, but performant defenses seem to be far away. In this work, we explore the use of edge-aware bilateral filtering as a projection back to the space of natural images. We show that bilateral filtering is an effective defense in multiple attack settings, where the strength of the adversary gradually increases. In the case of adversary who has no knowledge of the defense, bilateral filtering can remove more than 90% of adversarial examples from a variety of different attacks. To evaluate against an adversary with complete knowledge of our defense, we adapt the bilateral filter as a trainable layer in a neural network and show that adding this layer makes ImageNet images significantly more robust to attacks. When trained under a framework of adversarial training, we show that the resulting model is hard to fool with even the best attack methods.

## 1 Introduction

Deep neural networks are known to be vulnerable to targeted perturbations added to benign inputs. The perturbed inputs, known as *adversarial examples*, can cause a classifier to output highly confident, but incorrect predictions. The majority of prior work has studied adversarial examples in the context of computer vision, where they pose the clearest threat. Small perturbations, imperceptible to humans, can be added to input images that cause a classifier to output false predictions. Because of the particular success of neural networks in computer vision, these models are being deployed in areas such as autonomous driving, facial recognition, and malware detection. Recent work has shown that these systems are vulnerable in the real world to adversarial examples (Evtimov et al., 2017), which makes the problem of resisting adversarial attacks a growing concern.

There have emerged two central lines of research for defending against adversarial examples. *Denoising* approaches attempt to remove the adversarial perturbations from the inputs as a preprocessing step. This is often done by filtering, or by projecting the input to a lower dimensional space that cannot represent high frequency perturbations (Samangouei et al., 2018; Shen et al., 2017). These methods often lead to high accuracy, even on difficult datasets like ImageNet. But it has been shown that an attacker with knowledge of the defense can successfully circumvent them (Athalye et al., 2018). On the other hand, *Adversarial training* methods use principles from robust optimization to train models which resist adversarial attacks. Under the adversarial training framework, adversarial examples are combined with the natural training set to increase the model's robustness to attacks. These methods are expensive, requiring many more training examples, and have not been shown to scale well to natural image datasets such as ImageNet.

This paper explores the utility of bilateral filtering as both a denoising defense and a useful addition to adversarial training. Bilateral filtering is a classic approach in computer vision for edge-aware smoothing. Because natural images are more likely piecewise-smooth while adversarial perturbations are less likely to be, we hypothesize that bilateral filtering would be able to filter out adversarial noises. Indeed, in experiments we found that with appropriate parameters, a plain bilateral filter can recover **99%** of the adversarial images so that a classifier can predict the original label.

Furthermore, we introduce BFNet: an end-to-end model incorporating bilateral filtering as a differentiable layer. With BFNet, it is possible to examine the performance of white-box attacks trying to

bypass our bilateral filtering defense. We show that BFNet is naturally robust to attacks from many such adversaries, greatly reducing the strength of both $L_\infty$ and $L_2$ attacks on the ImageNet dataset.

Finally, we combine bilateral filtering with adversarial training, and achieve state-of-the-art results on MNIST and CIFAR10. Our method works with zero knowledge of either the network or any incoming attack, making it applicable to a variety of models and datasets.

## 2 RELATED WORK

### 2.1 ADVERSARIAL ATTACKS

There have been many proposed attacks for creating adversarial examples. We give a brief description of the six attacks that we used to test our models.

**A. Projected Gradient Descent (PGD)** In (Lyu et al., 2015; Madry et al., 2018), generating an adversarial example is the task of solving the objective $\max_{\delta \leq \epsilon} L(\theta, x + \delta, y_{true})$. PGD is used to maximize this objective under a loss function $L$, yielding an image with a perturbation magnitude less than $\epsilon$ with respect to the $L_\infty$ norm, and achieves the highest possible loss on the true class.

**B. Fast Gradient Sign Method (FGSM)** FGSM (Goodfellow et al., 2015) is a one step linearization of the above objective. FGSM finds adversarial examples by assuming linearity at the decision boundary. Given an image x, we find a perturbation $\eta$ under the max norm: $\eta = \epsilon \cdot sign(\nabla_x L(\theta, x, y))$, where $\theta$ is the parameters of the network, y is the original label, and $L$ is the loss function used to train the network.

**C. Momentum Iterative Method** The Momentum Iterative Fast Sign Gradient Method (MI-FGSM) (Dong et al., 2018) is an iterative version of the FGSM attack. MI-FGSM moves pixel values linearly along the gradient toward the decision boundary. MI-FGSM improves on FGSM by introducing a momentum term into gradient calculation: $g_{t+1} = \mu \cdot g_t + \frac{\nabla_x L(x_t^*, y)}{\|\nabla_x L(x_t^*, y)\|_1}$. The gradient is then used to iteratively update the image $x_{t+1}^* = x_t^* + \alpha \cdot (g_{t+1})$. The authors claim that simply using an iterative FGSM leads to greedy overfitting of the decision boundary, and thus falls into local poor maxima. Adding momentum stabilizes the update direction and creates a stronger adversarial example.

**D. L-BFGS-B** (Szegedy et al., 2014) used box-constrained L-BFGS to generate adversarial examples with minimal distortion under the $L_2$ norm. Given a natural image $x$ and a target class $y_{true}$, the adversarial objective is as follows: $\min \left[ c \cdot ||x - (x + \delta)||_2^2 + L(x + \delta, y_{target}) \right]$. Where $\delta$ is the adversarial perturbation, $L$ is the loss function, and the parameter $c$ controls the trade-off between the magnitude and strength of the perturbation.

**E. Carlini & Wagner Attack ($L_2$)** (Carlini & Wagner, 2017) proposed three iterative attacks which create adversarial examples under the $L_0$, $L_2$, and $L_\infty$ norms. In this work we consider the most powerful attack, the white-box $L_2$ attack. Specifically, they minimize $\min ||\frac{1}{2}(\tanh(w) + 1) - x||_2^2 + c \times f(\frac{1}{2}(\tanh(w) + 1))$, where $f(x') = \max(\max\{Z_i(x') : i \neq t\} - Z_t(x'), -\kappa)$. Here, $t$ is the target label, $Z$ refers to the logits of the network, $\kappa$ controls the confidence of the new classification, and the $\frac{1}{2} \tanh$ term constrains the result to pixel space.

**F. DeepFool** Deepfool is an iterative, first order method used to find minimal distortion under the $L_2$ norm (Moosavi-Dezfooli et al., 2016). Deepfool linearizes the classifier itself and performs gradient descent until the image is misclassified. The DeepFool objective is $\min_\delta \|\delta\|_2$ subject to $\arg \max f(x) \neq \arg \max f(x + \delta)$. In addition to the attacks listed above, other methods have been proposed. $L_0$ attacks such as (Papernot et al., 2016b) choose to measure adversarial perturbations by the minimum change necessary to produce an incorrect prediction.

### 2.2 ADVERSARIAL DEFENSES

There is a growing body of work on defenses against adversarial attacks (Papernot & McDaniel, 2016; Papernot et al., 2015; Xu et al., 2017; Liao et al., 2018). An averaging filter was studied in (Li & Li, 2017). JPEG compression was studied in (Dziugaite et al., 2016; Das et al., 2017), and was found to be effective at removing adversarial perturbations. However, JPEG encoding is not differentiable, hence its performance when the adversary has knowledge of the defense is unknown. Our bilateral filtering approach is fully differentiable hence we can test it against counter-attacks.

Other recent defenses attempt to remove adversarial perturbations by projecting inputs back onto the real data manifold (Meng & Chen, 2017). (Shen et al., 2017) projects inputs using a generative

adversarial network. Given a normal or adversarial image, a generator is trained to produce a image from the normal data distribution. This method also did not test against counter-attacks, and has been shown to be successfully fooled by the CW attack (Meng & Chen, 2017). Our approach can also be seen as a projection back to the data manifold, where we impose the constraint that the resulting image must be piecewise-smooth. By fixing the filter approach, we would likely not overfit significantly to the training set and remain effective under counter-attacks.

On the other hand, adversarial training methods (Goodfellow et al., 2015; Madry et al., 2018; Shaham et al., 2018; Tramèr et al., 2018) combine adversarial examples with the natural training set to increase the robustness of the model to adversarial attacks. These approaches are promising as they attempt to provide a guarantee on both the type of adversary and the magnitude of the perturbation they are resistant to. In practice however, these methods are hard to scale as they require expensive computation in the inner training loop to generate adversarial examples. When training on a large dataset such as ImageNet, generating a sufficient amount of strong adversarial examples can be intractable. This problem has been mitigated by training against a weak adversary like FGSM (Tramèr et al., 2018) which can quickly generate adversarial examples. But training models that are robust to strong adversaries on ImageNet or CIFAR-10 is still an open problem.

## 3 METHOD

In this paper, we consider white-box threat models where the attacker has full access to the training data, model parameters and architecture. This is categorically more difficult than black-box threat models where the attacker has little or no knowledge about the model or training data. We will first show the utility of bilateral filter against simple attacks without knowledge of the network, then introduce BFNet with bilateral filtering as a differentiable layer, so that we can evaluate attacks with knowledge of our defense.

### 3.1 THE BILATERAL FILTER AND ITS CAPABILITY OF RECOVERING ADVERSARIAL IMAGES

The bilateral filter is a non-linear Gaussian filter that is commonly used to smooth image gradients while preserving sharp edges. For an image $I$, window $\Omega$ centered at pixel $p$, the bilateral filter is formulated as a domain function $G_s$, and a range function $G_r$:

$$I_{filtered}(p) = \frac{1}{W_p} \sum_{q \in \Omega} G_s(\|\mathbf{p} - \mathbf{q}\|) G_r(\|I_p - I_q\|) \, I_q$$

where the normalization term $W_p$ is:

$$W_p = \sum_{q \in \Omega} G_s(\|\mathbf{p} - \mathbf{q}\|) G_r(\|I_p - I_q\|),$$

$G_s(x) = \exp(-\frac{x^2}{2\sigma_s^2})$ and $G_r(x) = \exp(-\frac{x^2}{2\sigma_r^2})$ are Gaussian filters, and $\sigma_s$ and $\sigma_r$ are parameters which control the strength of the domain and range functions respectively. Each neighboring pixel is assigned a weight according to both spatial closeness and value difference. Hence, if the color of the pixels $p$ and $q$ are very different, then $q$ will affect the filtered image at pixel $p$ very little. At sharp image boundaries, this would effectively lead to smoothing on only one side of the boundary, since the other side would have very different color. Hence, sharp boundaries can be preserved and oversmoothing or blurring that are commonly seen in Gaussian smoothing or averaging can be prevented. In Fig.1 one can see the effect of denoising an L-BFGS-B adversarial image, where an averaging filter will leave the image significantly blurred, but bilateral filtering would preserve the edges. More images are shown in the appendix in Fig. 3.

We believe piecewise-smoothness is an inherent property of many images hence bilateral filtering offered a projection back to this manifold of piecewise-smooth images. Convolutional networks only work on images from the natural image manifold, which left the hole for adversarial examples to maneuver by creating off-manifold images. By using bilateral filtering to force images to be on the manifold, we would leave significantly less holes for adversarials to maneuver on.

To test the efficacy of the bilateral filter to recover clean inputs from adversarial examples, we generated a set of adversarial examples from a range of powerful adversaries. Our first approach

was to manually tune parameters for each input image, to test the effective range of parameters which could recover the original label from an adversarial example. We found that with carefully chosen parameters, the corrupted labels could indeed be recovered. Our experiments showed that the small perturbations created by iterative methods like the Carlini & Wagner attack and DeepFool were easier to remove with a bilateral filter than the larger perturbations created with one step attacks. To remove perturbations generated by iterative attacks, we used small kernels 3 - 5 pixels wide, and $\sigma_s$, $\sigma_r$ values of $0.5$. Filtering with larger kernel sizes offers no benefit, as the resulting images from iterative attacks have imperceptible perturbations which are removed with small filters. One step attacks perturb every pixel in the image with the same magnitude of noise. As a result, we increased kernel width to 7 and $\sigma_s$ to 3, holding $\sigma_r$ constant. These parameters reliably removed adversarial perturbations from $L_\infty$ attacks with a bounded distance of $0.3$, as well as unbounded $L_2$ attacks. The results can be found in Table 1.

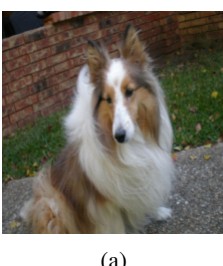 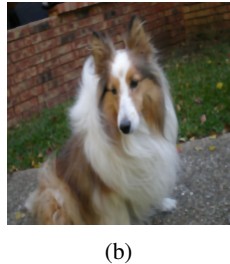 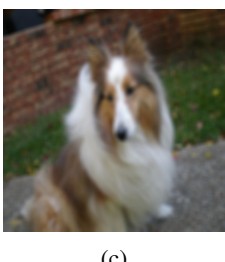

(a)           (b)           (c)

Figure 1: (a) The original LBFGS-B adversarial image, (b) The image after 3x3 bilateral filtering and (c) The image after 3x3 averaging filtering. The bilateral filter is superior since it removes small perturbations while preserving sharp edges in the image, keeping it from becoming blurry

| Network | FGSM | MI-FGSM | DeepFool | CW ($L_2$) | L-BFGS |
|---|---|---|---|---|---|
| Inception V3 | 97.0 | 97.5 | 98.8 | 99.2 | 97.8 |
| InceptionResNet V2 | 94.2 | 98.4 | 96.3 | 98.8 | 95.1 |
| ResNet V2 | 96.5 | 98.0 | 96.1 | 98.1 | 98.0 |

Table 1: Recovery performance for manually chosen bilateral filter parameters. We measure recovery by the percentage of examples which, after filtering, revert to the classification label assigned to the unperturbed image by the CNN. This shows that with adaptively chosen parameters according to the attack, we can recover nearly all adversarial examples

## 3.2 Adaptive Filtering

One caveat to the above approach, is that the parameters for the bilateral filter must be carefully chosen to be able to recover the accuracy and confidence of the original classification. Large values for the parameters $\sigma_s$ and $\sigma_r$ can create an excessively blurred image, and a small filter size $K$ may capture insufficient information to remove the adversarial perturbations. With this in mind, we train a small network which will predict the parameters of the bilateral filter $(K, \sigma_s, \sigma_r)$ for an input image. This network will serve as a cheap preprocessing step that will remove adversarial perturbations without affecting the underlying class label.

To build our classifier we first extract information about the distribution of pixel gradients by convolving the input with a Sobel filter in the $x$ and $y$ direction. Because adversarial attacks directly change values of the input, adversarial examples will often have larger color gradients in the $x$ and $y$ direction than natural images. We concatenate the gradient map depth-wise with the input image, and use three dilated convolutional layers with 64, 128, and 256 filters respectively, followed by 2x2 max pooling and a linear layer of 64 units. We use a dilation rate of 2 for each convolutional layer. Note this experiment is stand-alone and it is not utilized in the BFNet proposed in the next section.

## 3.3 BFNet: Adding Bilateral Filtering as A Trainable Layer

The main idea of BFNet is to always preprocess the input image with bilateral filter before inputting it into the CNN. Namely, instead of computing $f(x)$ where $f$ is learned by a deep network, always computing $f(BF(x))$ instead. Hence we can then optimize for attacks that have full knowledge and

gradients about our defense. This has two utilities, one is to examine the robustness of the defense, and secondly we can add the newly generated adversarial examples back to the training set of the network, in order to perform adversarial training.

A brute-force implementation of the bilateral filter has a $\mathcal{O}(n^2)$ cost associated with computing the response of individual pixels. Making it the most expensive operation in the graph. To reduce computation time, We choose as our preprocessing function the Permutohedral Lattice implementation of the bilateral filter (Adams et al., 2010), which is also fully differentiable and can be computed in $\mathcal{O}(n)$ time. This can then be attached as the first layer to any other network, and the bilateral filter parameters can be trained jointly with other parts of the network.

### 3.4 ADVERSARIAL TRAINING

It has been shown that under the white-box threat model, using a denoiser as the only defense is insufficient to stop the strongest adversarial attacks. Currently the most promising direction for training models robust to adversarial attacks is adversarial training. Despite continuing progress on both MNIST and CIFAR10, adversarial training is still very expensive, and performs worse than denoising approaches on the same datasets. We propose an approach combining adversarial training with BFNet, giving a robust, performant classifier on different threat models.

Following (Madry et al., 2018; Athalye et al., 2018; Carlini & Wagner, 2017), the adversarial training framework can be expressed as the following saddle point problem with model parameters $\theta$, and input $x$ with true label $y$:

$$\min_\theta f(x;\theta) \quad \text{where} \quad f(x;\theta) = \mathbb{E}_{(x,y)\sim D}\left[\max_\delta L(\theta, x+\delta, y)\right]$$

where a solution to the inner maximization problem represents the *most* adversarial example within some perturbation budget. Solving the outer minimization problem yields a classifier which is robust to the above adversary. (Madry et al., 2018) showed that PGD could reliably solve the inner maximization problem without linearization, and is thus a better adversary to train against than FGSM.

We propose a modification to the above saddle point formulation which incorporates the BFNet:

$$\min_\theta f(BF(x)) \quad \text{where} \quad f(x) = \mathbb{E}_{(x,y)\sim D}\left[\max_\delta L(\theta, x+\delta, y_{true})\right]$$

where $BF(x)$ is the bilateral filter in BFNet. Incorporating this, we train the entire BFNet adversarially, including the filtering layer. We report the results of our experiemnts in the following section.

## 4 EXPERIMENTS

### 4.1 ADAPTIVE FILTERING MODEL

In this section we show that our adaptive filtering model can correctly predict filtering parameters which will restore an adversarial input. To test this, we generate a dataset of 1,000 adversarial images from the ILSVRC 2012 validation set with five different attacks: Projected Gradient Descent with 40 steps (PGD), Box constrained L-BFGS, The Carlini & Wagner $L_2$ attack (CW), The Momentum Iterative FGSM (MIM), FGSM, and DeepFool. Where applicable, we constrain the perturbations to an $\epsilon$-ball of radius 0.3 from the training example. Source images have been normalized to a range of $[-1, 1]$.

To construct our training set we use a separate 1,000 images generated from each of the attacks in table 1. For each image, we collect labels in the form of triples $(K, \sigma_s, \sigma_r)$, $K$ denotes the kernel size, and $\sigma_s$, $\sigma_r$ are the standard deviation for the spatial and range kernels respectively. Given any adversarial example, there may be many permutations of parameters for the bilateral filter that successfully denoise the input. For this reason we collect a maximum of 10 different parameter configurations for each image in our training set. Given this is a multi-class prediction problem, we train using a sigmoid function at the output of our network to predict a set of candidate parameter configurations. At test time we evaluate with the parameters predicted by the maximally activated output unit. We evaluate results under two threat models. The first (presented in table 2) is where we attempt to change the label given to the adversarial example by our classifier. In this setting, we are

trying to defeat an attack who needs a *specific* label to be the output. In other settings, the attacker believes any label is sufficient so long as its not the true label. In this case we measure our ability to recover the true label in table 3.

We used the pretrained Inception V3 (Szegedy et al., 2016) and InceptionResNet V2 (Szegedy et al., 2017) ImageNet classifiers as our source networks. To generate adversarial examples on these networks, we used the open-source Cleverhans toolbox (Papernot et al., 2016a). the model was trained using SGD with Nesterov momentum for 25 epochs. We then test on six different validation sets, one for each adversary respectively.

| Source Network | Clean | FGSM | PGD | MIM | CW | DeepFool | L-BFGS |
|---|---|---|---|---|---|---|---|
| AF+Inception V3$_A$ | 95.0 | 89.0 | 90.7 | 79.1 | 89.1 | 90.3 | 81.3 |
| AF+Inception V3$_B$ | 95.0 | 95.9 | 98.0 | 96.4 | 94.1 | 95.3 | 96.2 |
| AF+InceptionResNet V2$_A$ | 91.1 | 87.2 | 87.1 | 75.3 | 87.8 | 85.0 | 80.8 |
| AF+InceptionResNet V2$_B$ | 91.1 | 93.1 | 98.0 | 94.3 | 97.8 | 92.0 | 95.3 |

Table 2: Performance of our adaptive bilateral filter (AF) network across different attacks. We show (A) the top-5 accuracy of recovering the original predicted classification label from the adversarial example (note this is not necessarily the ground truth label), as well as (B) how often AF is able to defeat the adversarial attack - changing the prediction from the adversarial label to a new one

| Network | Clean | FGSM | PGD | MIM | CW | DeepFool | L-BFGS |
|---|---|---|---|---|---|---|---|
| Inc V3 $_{top1}$ | 78.8 | 30.1 | 0.2 | 0.1 | 0.1 | 0.7 | 0.0 |
| Inc V3 $_{top5}$ | 94.4 | 65.2 | 4.8 | 5.5 | 7.3 | 0.5 | 12.1 |
| AF + Inc V3 $_{top1}$ | 71.7 | 71.0 | 71.6 | 63.1 | 71.1 | 70.1 | 64.2 |
| AF + Inc V3 $_{top5}$ | 89.6 | 84.0 | 86.3 | 74.6 | 84.1 | 85.2 | 76.7 |
| IncResNet V2 $_{top1}$ | 80.4 | 55.3 | 0.8 | 0.5 | 0.3 | 2.5 | 0.0 |
| IncResNet V2 $_{top5}$ | 95.3 | 72.1 | 15.8 | 10.2 | 10.3 | 8.5 | 19.2 |
| AF + IncResNet V2 $_{top1}$ | 73.1 | 70.1 | 70.8 | 60.5 | 70.3 | 70.5 | 65.0 |
| AF + IncResNet V2 $_{top5}$ | 86.7 | 83.1 | 82.8 | 71.7 | 83.6 | 85.6 | 77.0 |

Table 3: Top-1 and top-5 accuracy of InceptionV3 and Inception-ResNetV2 on adversarial examples. We can see that adaptive filtering significantly increases the robustness of the classifier against many diverse attacks

It can be seen that we recover adversarial examples generated by FGSM, PGD, CW and DeepFool near perfectly, while missing nearly 15% of the examples of MIM and L-BFGS. These results are significantly better than the results in (Li & Li, 2017), which used a 3x3 average filter to recover images. Our Adaptive Filtering network succeeds in removing adversarial examples generated on natural images, with a relatively simple network. This makes the Adaptive Filtering network a viable method for defending networks against an adversary who employs a wide range of attacks.

## 4.2 BFNet Defending Against Counter Attacks on ImageNet

Due to the high cost of adversarial training on natural images, on ImageNet we perform only one round of counter-attack, which is: have the attack knows about BFNet and attack it by backpropagating through the entire BFNet defense. We use this as an opportunity to test the robustness of BFNet that cannot be attributed to adversarial training. To this end, we use the Inception V3 and the Inception-ResNet V2 networks, and add our bilateral filter layer to the input, keeping the pretrained ImageNet weights. We test against both $L_2$ and $L_\infty$ adversaries to obtain a complete picture of the robustness of BFNet. $L_\infty$ is a more informative metric when discussing the magnitude of adversarial attacks on very small images, because a large perturbation measured under the $L_\infty$ norm equates to a large visual change across few pixels.

To measure resistance to attacks under the $L_2$ norm, we use the unbounded attacks L-BFGS and DeepFool. It is impossible to be fully resistant to unbounded attacks, because any image can be changed to a completely different image and its CNN output would certainly change. Hence, we report the average $L_2$ and $L_\infty$ distance of the adversarial images to the original ones from the unbounded attacks. From Table 4 we can see that our approach yields a very robust model against adversarial perturbations under the $L_2$ metric. When attacking our BFNet models with DeepFool,

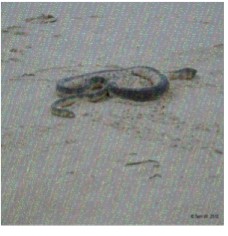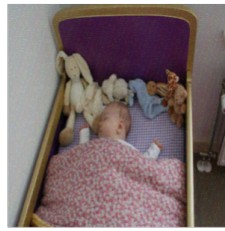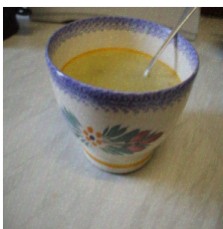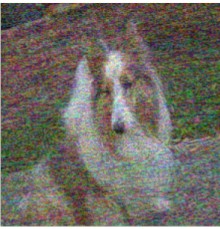

(a) Adversarial examples generated by L-BFGS on a BFNet version of the Inception V3 classifier. Generated adversarial examples have visually identifiable perturbations, and have an average $L_2$ norm of 106.2

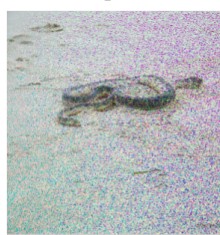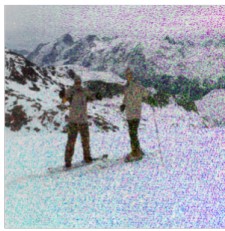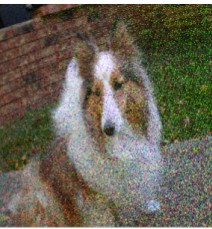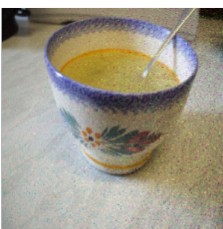

(b) Adversarial examples generated by DeepFool on a BFNet version of the Inception V3 classifier. The generated adversarial examples have large, noisy perturbations, and have an average $L_2$ norm of 181.2

Figure 2: Adversarial images created with BFNet. See the appendix for adversarial images from the same original images without BFNet

| Network | DeepFool | | L-BFGS | |
|---|---|---|---|---|
| | $L_\infty$ | $L_2$ | $L_\infty$ | $L_2$ |
| Inception V3$_{\text{Natural}}$ | 0.015 | 0.43 | 0.02 | 0.67 |
| Inception V3$_{\text{BFNet}}$ | 0.621 | 148.29 | 0.39 | 90.52 |
| IncResNet V2$_{\text{Natural}}$ | 0.025 | 0.44 | 0.06 | 0.77 |
| IncResNet V2$_{\text{BFNet}}$ | 0.793 | 187.45 | 0.65 | 90.65 |

Table 4: Performance of BFNet against DeepFool and L-BFGS attacks. We report the average $L_2$ and $L_\infty$ distance of $1,000$ adversarial images on Inception V3 and Inception-ResNet V2.

we see that the generated adversarial image has an $L_\infty$ distance over *30x larger*, when compared to an unmodified network of the same architecture. Similarly, we can see that the $L_2$ distance of an adversarial generated against BFNet is far larger when compared to adversarial images generated against a network of the same architecture without the bilateral filter. With respect to the L-BFGS attack, we see a similarly large disparity between BFNet and a vanilla network. Fig.2 shows some examples of images generated by those adversarial counterattacks. One can see that the DeepFool and LBFGS attacks had to significantly modify the image to defeat BFNet, creating clearly visible patterns.

For the $L_\infty$ attacks such as FGSM, and MI-FGSM we measure the resistance of our model to different values of perturbation $\epsilon$. We can see that our BFNet significantly decreases the attack strength of $L_\infty$ adversaries, in most cases by over $50\%$. Of particular note is that we show more significant resistance to adversarial perturbations of $\epsilon \leq 0.3$. Larger perturbations are visually discernible, and thus are less adversarial than smaller fooling perturbations. For both attacks we use 1,000 random images sampled from the ILSVRC 2012 validation set, and report the percentage of successful attacks against the natural model and BFNet respectively.

### 4.3 ADVERSARIAL TRAINING

Finally, we experiment on adversarial training with BFNet on the MNIST and CIFAR-10 datasets.

#### 4.3.1 MNIST

To show that BFNet is robust to strong first-order adversaries, we train a small CNN to $99.2\%$ accuracy on the test set. Our model consists of 2 convolutional layers with 32 and 64 filters respectively,

|  |  | FGSM | | MI_FGSM | |
| Network | Epsilon | Natural | BFNet | Natural | BFNet |
|---|---|---|---|---|---|
| Inception V3 | 0.1 | 26.8 | **30.2** | 58.8 | **21.0** |
| Inception V3 | 0.15 | 21.4 | **36.6** | 65.6 | **30.1** |
| Inception V3 | 0.3 | 6.8 | **46.6** | 88.2 | **42.2** |
| Inception V3 | 0.5 | 1.0 | **63.2** | 98.0 | **52.4** |
| Inception V3 | 0.75 | 0.0 | **90.8** | 99.6 | **53.4** |
| Inception V3 | 1.0 | 0.0 | **99.8** | 99.6 | **70.8** |
| IncResNet V2 | 0.1 | 41.2 | **42.6** | 89.1 | **59.0** |
| IncResNet V2 | 0.15 | 34.4 | **55.4** | 90.6 | **38.6** |
| IncResNet V2 | 0.3 | 11.8 | **75.4** | 92.2 | **59.4** |
| IncResNet V2 | 0.5 | 2.0 | **91.0** | 100.0 | **74.6** |
| IncResNet V2 | 0.75 | **0.4** | **99.6** | 100.0 | **80.2** |
| IncResNet V2 | 1.0 | **0.4** | **99.6** | 100.0 | **91.8** |

Table 5: Performance of BFNet against FGSM and MI-FGSM adversaries for a range of perturbation sizes (lower is better). For our MI-FGSM attack, we use a momentum decay factor of $1.0$, and run the attack for $10$ iterations

each followed by 2 x 2 max pooling and ReLU. We use a final fully connected layer with 1024 units. We modify our network into a BFNet by adding our bilateral filter layer at the input of the first convolutional layer. We then train the entire BFNet (including the filtering layer) with adversarial training using three distinct adversaries: FGSM, PGD, and PGD with the proposed CW loss function. We report the results in table 6. Our results perform well against the state-of-the-art adversarial training results. We also show that when our network is trained on a single strong adversary, we are robust to attacks from other adversaries.

| Network | Clean | FGSM | PGD | CW | CW ($\kappa$=50) | Method | Type | Clean | FGSM | PGD |
|---|---|---|---|---|---|---|---|---|---|---|
| BFNet$_{pgd}$ | **99.0** | 95.5 | **98.0** | 93.2 | - | BFNet$_{pgd}$ | A | 87.1 | 55.2 | **50.4** |
| BFNet$_{fgsm}$ | **99.0** | **98.1** | 36.4 | 88.2 | **96.0** | BFNet$_{pgd}$ | B | 73.1 | 64.5 | 38.1 |
| Madry | 98.8 | 95.6 | 93.2 | **94.0** | 93.9 | BFNet$_{fgsm}$ | B | 76.5 | **70.6** | 12.2 |
| Tramer$_A$ | 98.8 | 95.4 | 96.4 | - | 95.7 | Madry | A | 87.3 | 56.1 | 45.6 |
| Tramer$_B$ | 98.8 | 97.8 | - | - | - | | | | | |

Table 6: LEFT: Comparison of our method with state of the art adversarial training results on MNIST. BFNet$_{pgd}$ denotes our model trained against a PGD adversary, while BFNet$_{fgsm}$ is trained against a FGSM adversary. For Tramer We report A: the strongest white box attack given against a non-ensembled model from (Tramèr et al., 2018), as well as B: the performance of architecture B from (Madry et al., 2018); RIGHT: Performance of our two adversarially trained BFNets on CIFAR-10. BF$_{pgd}$ denotes our model trained against a PGD adversary, while BFNet$_{fgsm}$ is trained against a FGSM adversary. Network type (A) refers to the ResNet network used in (Madry et al., 2018), while (B) refers to the smaller architecture.

During training we observe a faster convergence in training loss (see Appendix), and increased robustness to white-box FGSM, PGD, and CW attacks, when trained against only the PGD attack. However, the model trained against FGSM does worse against stronger adversaries such as PGD, as the attack itself is a weak adversary.

### 4.3.2 CIFAR10

We perform similar experiments to test BFNet on CIFAR-10. We use a network with four convolutional layers, each followed by 2x2 max pooling. A linear layer of $4,096$ units is used before softmax. For BFNet, a differentiable bilateral filter layer to preprocess the images. When naturally trained with Adam for 30 epochs, this network reaches an accuracy of 79.04% on the test set. We also train the original ResNet-18 model used in (Madry et al., 2018) for 80K iterations. Trained on natural examples we reached an accuracy of 92.7% on the test set. Each model is then trained adversarially with PGD and FGSM, respectively. We use an $L_\infty$ bound of $\epsilon = 8$ for both adversaries. We

use 20-step PGD with a learning rate of 2.0. We report our results in Table 6 and it can be seen that our BFNet trained on PGD outperforms (Madry et al., 2018) significantly on the PGD adversary.

In contrast to MNIST, CIFAR-10 remains a very challenging dataset. The higher dimensionality makes robust training significantly more difficult. Because CIFAR-10 is too small, the edges are not so obvious, which could have hurted our performance. We believe that the bilateral filtering poses more constraint in larger natural images, such as ImageNet.

## CONCLUSION

The continued existence of adversarial examples, and the lack of effective defenses limits our ability to deploy AI systems in critical areas where safety and security are necessary. In this paper we showed that a bilateral filter can be used as a core part of versatile, effective defenses to recover clean images from perturbed ones. The bilateral filter remains effective when deployed with numerous defense strategies: as a manual preprocessing step, a trained denoiser, or a robust model that is trained end-to-end. Our defense holds in multiple attack settings where the attacker has knowledge about it. In the future we hope to explore even better filter approaches which projects even better to the natural image manifold and limit adversarial examples, as well as combining it with an adversarial detection approach to construct a comprehensive defense.

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

# 5 APPENDIX

## 5.1 MORE EXAMPLES OF BILATERAL FILTERING RESULTS ON ADVERSARIAL IMAGES

Fig. 3 shows more examples of adversarial images from several different attacks after bilateral filtering.

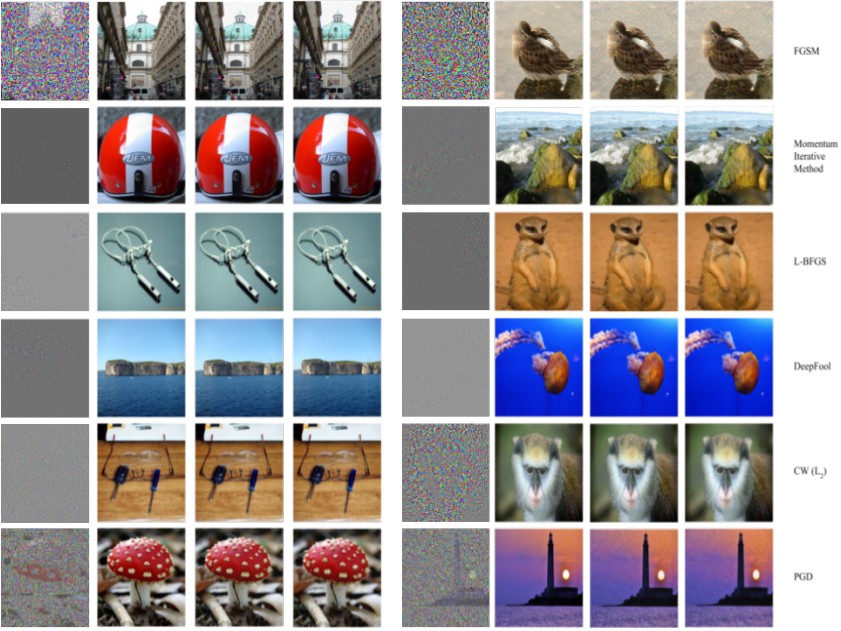

Figure 3: The effect of bilateral filtering on adversarial inputs. From left to right, we show the adversarial perturbation, the clean image, the adversarial images generated by the respective attack algorithms, and the recovered image after bilateral filtering. Note that bilateral filtering does not destroy image quality, and images can be correctly classified.

## 5.2 ADVERSARIAL IMAGES WITHOUT BFNET

Fig. 4 shows the images generated with DeepFool and L-BFGS on ImageNet without BFNet added to preprocess the images. Compared with Fig. 2.

## 5.3 CONVERGENCE OF THE ADVERSARIAL TRAINING

Fig. 5 shows the convergence of the adversarial training.

## 5.4 ADVERSARIAL EXAMPLES THAT FOOLED THE ADVERSARIALLY TRAINED BFNET

Fig.6 and Fig.7 showed all the failure images after adversarial training on the MNIST dataset.

## 5.5 BLACK-BOX ATTACKS ON BFNET

We performed two seperate black box attacks on BFNet to tests its robustness to a new kind of adversary. We tested on the non-adversarially trained imagenet models InceptionResNet-V2 and Inception-V3 (corresponding to Table 4). We report the results in the updated appendix of the paper. For both attacks we used the implementation given in the Foolbox toolbox.

First, we used the Boundary attack from (Brendel et al., 2017): a decision-based attack that begins with a large adversarial perturbation, then reduces the magnitude of the perturbation while retaining an adversarial label. This attack has been shown to be effective and scale to high dimensional inputs.

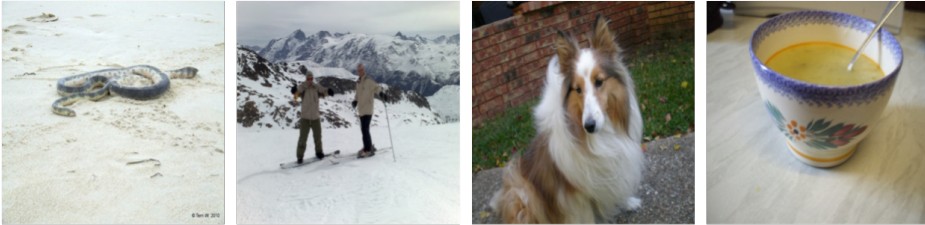

(a) Adversarial images generated with DeepFool on a vanilla Inception V3 classifier. The adversarial images are visually identical to the real images, and have an average $L_2$ norm of 0.09

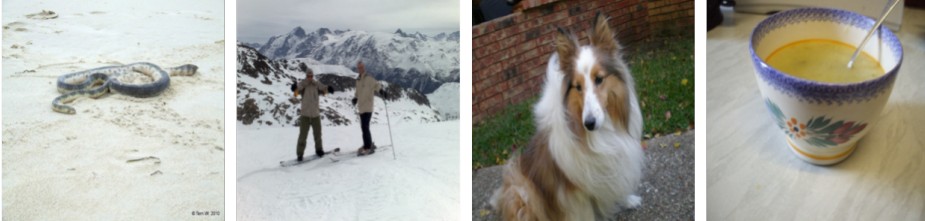

(b) Adversarial examples generated by an L-BFGS adversary on a vanilla Inception V3 classifier. The adversarial examples have an average $L_2$ distance of 0.025 from their natural counterparts, and have visually imperceptible perturbations.

Figure 4: Adversarial images generated by DeepFool and L-BFGS without BFNet, to be compared with Fig.2

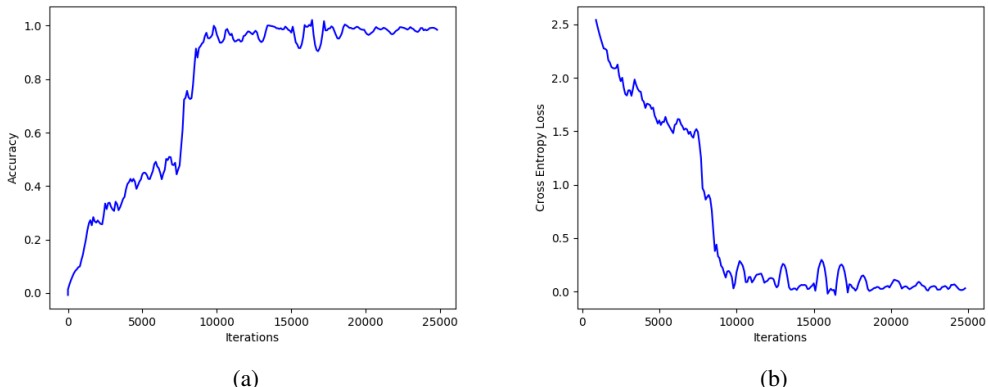

(a)                       (b)

Figure 5: The average mini-batch batch accuracy (a) and cross entropy loss (b) for the model trained with adversarial training on MNIST. We trained our model to convergence, which happened near 20k iterations. We can see that the training is stable and converges to a similar training error as a naturally trained network

Second, we used an approximate L-BFGS attack where the gradients are computed numerically. Using a numerical gradient avoids the instabilities in iterative attacks which occur in gradient-masking defenses. We report the average L2 distance achieved of these attacks against BFNet of 1000 images. We show the results in table 7 From the results given, we can see the BFNet does well against both the decision-based attack and the approximate L-BFGS, showing that it is indeed resistant to such kind of attacks as well.

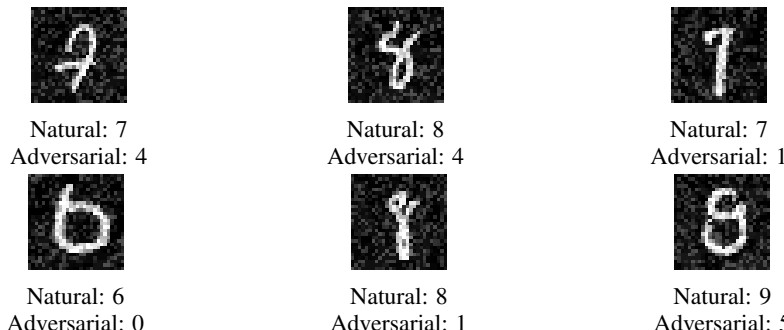

Figure 6: PGD adversarial examples which fool an adversarially trained BF$_{\text{PGD}}$ with $\epsilon = 0.3$

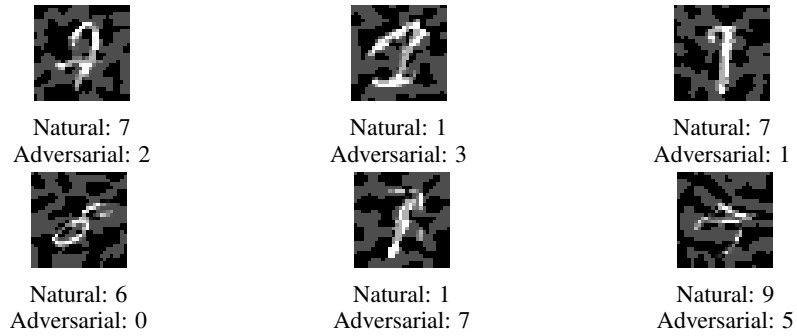

Figure 7: FGSM adversarial examples which fool adversarially trained BF$_{\text{fgsm}}$ with $\epsilon = 0.3$

| Network | Boundary-Attack $L_2$ | Approx. L-BFGS $L_2$ |
|---|---|---|
| Inception V3$_{\text{Natural}}$ | 7.20 | 0.82 |
| Inception V3$_{\text{BFNet}}$ | 52.562 | 50.21 |
| IncResNet V2$_{\text{Natural}}$ | 8,74 | 1.34 |
| IncResNet V2$_{\text{BFNet}}$ | 30.33 | 34.61 |

Table 7: Performance of BFNet against the Boundary and Approximate L-BFGS black-box attacks. We report the average $L_2$ distance of $1,000$ adversarial images on Inception V3 and Inception-ResNet V2.

