# OpenReview forum: "Unifying Bilateral Filtering and Adversarial Training for Robust Neural Networks"
_ICLR.cc/2019/Conference_

### Official Review · AnonReviewer2 · 2018-10-30
**Distortion and Obfuscated Gradients**

**Rating:** 5
**Confidence:** 3

**Review:**

The paper addresses the robustness of deep neural networks to adversarial example attacks. It uses a bilateral filtering as a preprocessing step to recover clean data from adversarial ones. It can also get combined with adversarial training and be trained end-to-end. The paper is well written, the background and introduction is clear. However I have comments about their implementation and experimental results:

1. They are claiming to have a very high distortion while their model can still perform well.  They have to make sure obfuscated gradients has not happened and they have implemented the back-propagation from attack to defense correctly.

2. Also if they had consider black-box threats as well as white-box one, it would have been more informative of how their method actually performs. Specially, in order to check whether obfuscated gradients has happened, the black-box threats are better choice, which they have not tried.

So considering the above, there might be an issue in what they are claiming, so they might reconsider their method. Maybe reviewing their codes by some experts in this topic can give a good evaluation.

(I have to mention that I am not an expert in Adversarial networks)

---

> ### Author Response · Authors · 2018-11-26
> **Reply**
>
> We appreciate the point brought forward by the BPDA paper, that gradient masking is often the cause of a defense that looks good on the outside yet is easily broken and agree with the reviewer that those experiments are needed.
>
> To this end, we performed two black box tests on the non-adversarially trained ImageNet models InceptionResNet-V2 and Inception-V3 (corresponding to Table 4). We report the results in the updated appendix of the paper. For both attacks, we used the implementation given in the Foolbox toolbox. We give the results of our experiments in table 7.
>
> First, we used the Boundary attack from [1]: a decision-based attack that begins with a large adversarial perturbation, then reduces the magnitude of the perturbation while retaining an adversarial label. This attack has been shown to be effective and scale to high dimensional inputs.
>
> Second, we used an approximate L-BFGS attack where the gradients are computed numerically. Using a numerical gradient avoids the instabilities in iterative attacks which occur in gradient-masking defenses.
>
> From the results given, we can see the BFNet does well against both the decision-based attack and the approximate L-BFGS, showing that it is indeed resistant to such kind of attacks as well.

---

### Official Review · AnonReviewer3 · 2018-11-03
**The paper is well written, however some experimental results are incomplete/unclear and discussion is lacking**

**Rating:** 5
**Confidence:** 5

**Review:**

1. The numbers in Table 1 should be compared with the %-age of examples that were actually fooled to begin with by using each adversarial attack scheme under study. Only then can the recovery performance of the hand-tuned bilateral filtering be put into perspective.
2. In case of the adaptive filtering method, the parameters for bilateral filtering are predicted by a standalone classifier. For the experiments to be fair, any incoming image, whether adversarially perturbed or not, must be filtered using the parameters predicted by the classifier. Thus, are the results in the column titled 'clean' pertaining to when natural (unperturbed images) are filtered and tested? If not, then what are the accuracy values in that case.
3. Also in Table 2, the case (B) measures the ability of the adaptive filtering to change the predicted label from the adversarial label to a 'new' one.  In this case, it is important to measure/verify whether the 'new' label is benign or malignant (which may well depend on the specific task of classification and the semantic associations of the output classes). Without this analysis for the 'new' class, it seems irrelevant and even misleading to quote this measure.
4. It is unclear what the relation between the results in Table 2 and Table 3 is. How are the values in the two tables linked, if at all?
5. In Table 3., once again, what does 'clean' column report? Are the natural images adaptively filtered and then tested for accuracy? If not, then do also report those numbers in a separate column.
6. It is really evident in Table 3 that as the adaptive filtering is applied and the robustness seems to improve (as measured in terms of the accuracy figures for the various attack schemes), the accuracy for natural/clean images consistently and significantly drops. There has not been discussed anywhere in the paper. What does this imply? It should be highlighted in text and maybe even graphically, plotting the tradeoff between robustness gain (on adversarial images) and accuracy loss (on clean images).
7. Some claims in the paper are unsubstantiated: (a) 2nd para, pg 7 - an adversarial detector or a human eye would be able to detect those attacks. Do the authors have any quantitative figures to share on how well a detector or human subjects would fare at this task? (b) Larger perturbations are visually discernible and thus less adversarial? Again, from the images in row 2, fig2, one can notice that there is a higher quotient of noise but this doesn't in any way interfere with the recognisability of those images by a human. How are they less adversarial when humans are not affected by the perturbations as concerns the classification task?
8. What does the column title 'natural' in Table 5 correspond to, especially as it is a subcolumn of an attack scheme? Also, the numbers in Table 5 seem to be the fooling rates. This is inconsistent with the remaining tables where accuracies are being presented. Do you think it would be better to convert these to accuracy figures too?

I am interested in knowing the authors' responses on the above points before I can propose my final rating.

---

> ### Author Response · Authors · 2018-11-26
> **Reply**
>
>
> We thank the reviewer for the concerns raised about our proposed method. We wish to address them below, as well as in the revision of our paper.
>
> Question 1: The numbers in Table 1 should be compared with the %-age of examples that were actually fooled.
> - 100% of the adversarial examples considered in Table 1 fooled the network. We have already excluded failed adversarial attacks beforehand. We have made this more clear in the revision.
>
> Question 2: Are the results in the column titled 'clean' pertaining to when natural (unperturbed images) are filtered and tested? If not, then what are the accuracy values in that case.
> - In table 3, the column labeled “clean” refers to the top1 and top5 accuracy of the networks with adaptive filtering, where each incoming image (including natural unperturbed ones) is filtered with the predicted parameters.
>
>
> Question 3-5: In Table 2-3 why does diversion to a new category make sense? What are the relationships between Table 2 and 3?
> - In table 2, we considered the threat model where an attacker wants an ML system to give a specific output. In this case, it is enough to destroy the targeted perturbations created by the attack to foil the attacker. In other cases (like autonomous vehicles) we are interested in the correct label above all else, we report results for recovering the correct label in table 3.
>
>
> Question 6: With adaptive filtering, the accuracy of natural/clean images consistently and significantly drops, why?
> - The accuracy drop for using adaptive filtering is due to adaptive filtering not being a powerful enough method to detect adversarial examples. It can reliably assign filtering parameters which may improve robustness, but this comes at a cost of potentially destroying discriminant features. The fact that adaptive filtering does not have the classification accuracy we expect is partially what leads us to a more end-to-end solution like BFNet.
>
>
> Question 7: How are larger perturbations less adversarial – especially when a human can still correctly classify the perturbed images?
> - We agree that notions of ‘adversarialness’ being tied to human perception are vague. We have removed these comments from the manuscript. It is although a common standard in most papers to measure adversarial examples based on the L_2 or L_infty norm of the perturbations, which we adopt as well.
>
> Question 8: The numbers of Table 5 is fooling rate which is inconsistent with every other table.
> - We have converted the natural column from error rate % to accuracy in the manuscript.

---

### Official Review · AnonReviewer1 · 2018-11-05
**An interesting idea, but not good enough. Fundamental flaws in the evaluation.**

**Rating:** 4
**Confidence:** 5

**Review:**

The paper proposes the use of bilateral filtering as a defense to adversarial examples (AE). Bilateral Filtering (BF) is a smoothening technique that averages pixels that are close to each other both in position and value. As a result BF preserves sharp edges and the denoised images appear to be of higher quality than those produces via simple Gaussian smoothing.
First, the method is evaluated on black-box attacks by constructing AE for the undefended network and then applying the defense. The method is then tested on white-box attacks which attack the defended system end-to-end with first order methods. Finally, the authors propose combining their defense with adversarial training.

Overall, I find the main idea of the paper interesting. BF seems like a natural approach to remove low-magnitude perturbations from the image while preserving the salient characteristics of the image. Thus, if it did not hurt classification accuracy significantly it might be considered as part of a cheap preprocessing pipeline that protects the model from simple attacks. However, as shown in Table 3 and 6, BF reduces the accuracy of the original model by at least 5% on ImageNet and CIFAR10.

My main objection about the paper is that BF is proposed as a defense that is extremely robust against even the best attacks. However, the evaluation is clearly incomplete for such a claim and fails to pass simple sanity checks.

*Epsilon values used to claim robustness are way too high*. Table 4 reports an L_infinity bound of 0.793 (almost 40% the image range) and L_2 of 187 (again roughly a quarter of the L2 radius of the image domain). One can change the class of the image completely (even for humans) with a smaller epsilon bound. This is a very clear indication that the attacks used are not sufficient to properly evaluate the robustness of the defense. Even in the images produced by the authors (Figure 2) it is clear that these perturbations are sufficient in terms of magnitude to completely distort the image.

*Only first-order attacks are considered*. From the point about epsilon values above, it is fairly clear that the defense is not easy to attack with first-order methods. This is understandable since BF, even if "fully differentiable", will lead to vanishing gradients for certain pixels. I think it is necessary that the authors to evaluate their defense on other attacks such as a) a straight-through approximation of BF (see BPDA from https://arxiv.org/abs/1802.00420), b) a finite-differences-based attack (see SPSA from https://arxiv.org/abs/1802.05666). As a sanity check, I would even recommend evaluating using the nearest neighbor of each image from a wrong class. It is clear that this attack will always succeed and the L_p distance will be smaller than those reported here.

I believe that these are fundamental issues about the paper that need to be addressed before even considering the paper for acceptance and I thus recommend rejection at this time.

Other comments and concerns:
--Table 6: what is the performance of BFNet without adversarial training? This is a necessary baseline to understand the results of the table.

-- I am confused by the adaptation of adversarial training to BFNet (last equation of section 3.4). The equations themselves are confusing since x is used twice (as a parameter to f and sampled from D). My first guess would be that the method is performing end-to-end adversarial training from the loss to the input x (before BF). It appears however that adversarial training is performed with respect to the image *after* applying the BF filter. If this is the case, it goes against the fundamental principles of adversarial training.

-- I do not see the point of adaptive filtering (AF) as a defense. BF is attractive because it is simple. Adding a classifier to figure out the right parameters seems dubious at best. The idea seems fundamentally flawed given that the classifier is only trained on known attacks and might thus not capture new, different attacks. Lastly, since the classifier is a ML model, it will clearly be susceptible to adversarial perturbations. So the adversary can choose an attack that fools the original model and forces the classifier to choose bad BF parameters. I would suggest that the authors completely remove AF from future versions of their paper.

-- The performance on ImageNet is significantly better than the performance on MNIST and CIFAR10. This is counter-intuitive since a much harder dataset both from a standard and adversarial point of view.

Minor comments to authors:
-- A JPEG compression-based defense can be bypassed (https://machine-learning-and-security.github.io/papers/mlsec17_paper_54.pdf).
-- Last paragraph of page 3 is too informal ("less holes for adversarials to maneuver on").
-- Top of page 4: you seem to confuse "iterative methods" with "small magnitude" attacks. One can use iterative attacks to create large adversarial perturbations (e.g. PGD).
-- I would suggest normalizing to the more common [0, 1] range as the current numbers can be confusing.
-- I am confused by Tables 2, 3. First row of Table 2 shows that on 95% of the clean inputs the label is not changed after BF. But table 3 shows that the accuracy drop from 78.8% to 71.7%. What is going on here?
-- Last part of first paragraph of 4.2 is hard to understand.
-- End of page 8, "the higher dimensionality of CIFAR10 makes it harder than MNIST". Yet the paper claims excellent performance on ImageNet.

---

> ### Author Response · Authors · 2018-11-26
> **Reply**
>
> We thank the reviewer for the comments and concerns raised here. We have made revisions to our manuscript, and we will answer outstanding questions here.
>
> Question: Epsilon values from Table 4 are way too high and the paper should consider finite-difference based attacks such as SPSA and other black box attacks.
>
> Answer: We appreciate the point brought forward by the BPDA paper, that gradient masking is often the cause of a defense that looks good on the outside yet is easily broken, and we agree with the reviewer that those experiments are needed.
>
> To this end, we performed two black box tests on the non-adversarially trained ImageNet models InceptionResNet-V2 and Inception-V3 (corresponding to Table 4). We report the results in the updated appendix of the paper. For both attacks, we used the implementation given in the Foolbox toolbox. The results are found in table 7.
>
> First, we used the Boundary attack from [1]: a decision-based attack that begins with a large adversarial perturbation, then reduces the magnitude of the perturbation while retaining an adversarial label. This attack has been shown to be effective and scale to high dimensional inputs.
> Second, we used an approximate L-BFGS attack where the gradients are computed numerically. Using a numerical gradient avoids the instabilities in iterative attacks which occur in gradient-masking defenses.
>
> From the results given, we can see the BFNet does well against both the decision-based attack and the approximate L-BFGS, showing that it is indeed resistant to such kind of attacks as well.
>
> Question: How is adversarial training adapted to BFNet (Sec. 3.4)? Is it applied to the image after applying the bilateral filter?
> Answer: Adversarial training is performed on the entire BFNet, including the bilateral filtering layer, which means the gradient goes through the bilateral filter and goes to the original image before the bilateral filtering layer. We have cleared the notations in the paper.
>
> Question: Adaptive filtering is a weak defense and should be removed
>  We used adaptive filtering as a motivation to BFNet -- showing that a plain BF with parameter selection can defeat many adversarial examples, but insufficient because it reduces accuracy and can be brittle. It is indeed less capable than the manual filtering, or BFNet. We are open to explicitly mention that or removing this part from the paper if there is a consensus that it’s not needed.
>
> Question: In Table 6, what is the performance of BFNet without adversarial training?
> - Without adversarial training, we can only fine-tune the parameters of the vanilla network to adapt to the bilateral filter. - At first, the accuracy is quite poor. But with fine tuning with 10 epochs, BFNet with both classifiers (InceptionResNet-V2 and Inception-V3) lose only 5% accuracy on the ImageNet test set. We will include this metric in the final paper.
>
> Question: Why are ImageNet results significantly better than MNIST and CIFAR?
> We believe the answer to this question is that bilateral filtering is a much more sound approach in high-resolution images such as ImageNet than low-resolution ones such as MNIST and CIFAR. In the 32x32 resolution in MNIST and CIFAR, a bilateral filter can only be very narrow to perform well, otherwise, it would easily distort the image significantly to kill the signal on the original class. On the other hand, with the high resolution in ImageNet, we can perform a significant amount of bilateral filtering (in terms of both the width and range) without destroying the content of the image, hence the defense effect would be significantly higher than in low-resolution images.
>
> [1] Brendel, Wieland, Jonas Rauber, and Matthias Bethge. "Decision-based adversarial attacks: Reliable attacks against black-box machine learning models." arXiv preprint arXiv:1712.04248 (2017).

---

> > ### Comment · AnonReviewer1 · 2018-11-26
> > **Boundary attack and Approx. BFGS reduce performance significantly**
> >
> > I appreciate the authors' response and the time spent performing additional experiments.
> >
> > If I understand correctly, Table 7 can be directly compared with Table 4. The difference between these tables is concerning. The boundary attack is able to find perturbations of average size *significantly* smaller than those found using previous attacks. Specifically, according to Table 4, the average perturbation required to fool Inceptionv2 is 90 where Table 7 reveals that it is below 30. A factor of 3 in perturbation size is significant. This is clear indication that the overall evaluation is unreliable at this point. Independently of the ICLR decision I would urge the authors to carefully revisit their evaluation process.

---

> > > ### Author Response · Authors · 2018-11-29
> > > **Black Box Evaluation**
> > >
> > > While the black box attacks that the reviewer suggested performed better than white-box ones, the results show that they were not able to shatter the BFNet defense. We do not believe that this experiment invalidates our method.
> > > Our evaluation result is in line both with the reviewer’s assumption (that black-box may do better than white-box) as well as our own assumptions (that BFNet is robust), we don’t see how this additional experiment “clearly indicates the overall evaluation is unreliable at this point”. We have been making honest and sincere efforts in all our experiments, including both the original ones and the additional ones.
> > >
> > > Table 4 and BFNet constitute only an intermediate part of our paper. Our main defense proposed in the paper is BFNet combined with adversarial training. The notion that BFNet may be broken by a stronger attack is something we already considered in section 3.4. This is precisely the reason why we employ adversarial training. We believe the consideration of this paper should not stop at Table 4.

---

> > > > ### Comment · AnonReviewer1 · 2018-11-29
> > > > **On evaluating adversarial robustness**
> > > >
> > > > I truly appreciate the authors responses and additional experiments performed. Let me elaborate on what I mean by "unreliable evaluation".
> > > >
> > > > When proposing a new adversarial defense, the goal is to produce a model that is robust against _all_ possible attacks within a threat model. For instance, increased robustness to FGSM attacks does not constitute progress if the model is vulnerable to PGD attacks. This is why the robust accuracy of a model is defined as the *minimum* accuracy achieved against the worst-case attack within the threat model.
> > > >
> > > > If I'm not mistaken, the goal of table 4 is to show how BFNet improves the robustness of the model. It is thus necessary that Table 4 is an accurate evaluation of the model's robustness. However, given that a new attack reduces the robustness of the model by *a considerable amount*, this implies that original evaluation was not accurate and the model is significantly weaker than originally claimed. As a result, we cannot consider Table 4 as evidence supporting the robustness of BFNet unless further evaluation is performed.
> > > >
> > > > The authors argue that the main contribution of their paper is the combination of BFNet and adversarial training. However the results here are very weak. According to Table 6, the robust accuracy on MNIST is 93.2% (against the CW attack) which is exactly on par with the "Madry" model. On CIFAR10, the proposed methods achieves robust accuracy of 50.4% while the "Madry" model 45.6%. This improvement is small and given how PGD failed to properly evaluate the robustness of a BFNet only defense I am skeptical about it's effectiveness here too.

---

> > > > > ### Author Response · Authors · 2018-12-01
> > > > > **Note on our results and evaluation**
> > > > >
> > > > > Thank you for your continued engagement with us on this matter.
> > > > >
> > > > > In the final version, we will merge Table 4 and Table 7 into one table, which will solve the validity issue we have discussed in detail above, since in Table 7 we did test against the best attacks, as requested. We would like to reiterate that BFNet is still significantly better than the ImageNet filtering defense [2], tested in [1]. Using the L2 normalization in [2], the numbers in our Table 7 translate to (0.091, 0.091), and  (0.055, 0.062) for Inception V3 and InceptionResNet V2 respectively. Which is significantly better than the considered filtering defenses, which failed under a perturbation of 0.01 against a BPDA adversary. Our approximate L-BFGS uses a finite-difference method to provide the gradient, which is a BPDA strategy to attack BFNet. Hence we are at least 5x - 6x more robust than the other analyzed ImageNet defenses. We will clarify this and provide the comparison using comparable numbers in the final version.
> > > > >
> > > > > From that result and our performances in adversarial training, we believe we are clearly better than state-of-the-art, which we believe should be achieving the standard for a paper to be published.
> > > > >
> > > > > [1] Athalye, Anish, Nicholas Carlini, and David Wagner. "Obfuscated gradients give a false sense of security: Circumventing defenses to adversarial examples." arXiv preprint arXiv:1802.00420 (2018).
> > > > >
> > > > > [2] Guo, C., Rana, M., Cisse, M., and van der Maaten, L. Countering adversarial images using input transformations. International Conference on Learning Representations, 2018.

---

### Public Comment · (anonymous) · 2018-10-24
**Black-box attacks?**


In your paper, the claim about white-box "This is categorically more difficult than black-box threat models where the attacker has little or no knowledge about the model or training data" may not be true. If a model causes obfuscated gradients, black-box attacks perform better than white-box attacks (https://arxiv.org/abs/1802.00420). How do you make sure if your model does not cause obfuscated gradients?

---

> ### Public Comment · (anonymous) · 2018-10-24
> **White box attacks are still more powerful**
>
> White box attacks have strictly more information available to them and therefore are more powerful by definition.
>
> The problem is that with gradient masking even though more information should only help, sometimes it hurt (indicating that the attack is performing sub-optimally). So it makes sense to run both white-box and black-box attacks and verify the white-box attacks do better.

---

### Public Comment · (anonymous) · 2018-10-25
**Concerns about Table 5**

It looks like you are considering epsilon values larger than 1/2. With this distortion, any image can be converted to a solid grey image, and accuracy should be 1/N% in the best possible case. It looks like you are still seeing robustness at this distortion level, indicating the attacks are performing sub-optimally. Or is there something else going on here?

---

> ### Public Comment · (anonymous) · 2018-10-25
> **Additional Comments**
>
> I agree with this opinion. If attacks are unbounded, a model should get 0% accuracy. Otherwise, the model causes obfuscated gradients.

---

> ### Author Response · Authors · 2018-10-29
> **Normalization Range**
>
> Hi there, thanks for considering our work. We normalized the images to the range [-1, 1] (we mention this in section 4.1) instead of [0, 1]. We did this because we used the Cleverhans library to generate adversarial examples, and most pre-trained TensorFlow models accept data in the [-1, 1] range. Here an epsilon value of 0.5 is only 25% of the effective range. Our method indeed sees degraded performance as the perturbation magnitude approaches 1.0 (half the effective range). At epsilon = 1.0, we expect to see the attack succeed because it destroys the image. Indeed this happens with FGSM, and to a lesser degree, MI-FGSM.

---

> > ### Public Comment · (anonymous) · 2018-10-30
> > **Cleverhans normalizes images to the range [0, 1]**
> >
> > In the Cleverhans tutorial code[1], images are normalized to the range [0, 1], not the range [-1, 1]. Maybe you need to check the influence of the adversarial perturbations in different normalization range.
> >
> >
> >
> > [1] https://github.com/tensorflow/cleverhans/blob/master/cleverhans_tutorials/mnist_tutorial_keras_tf.py (code line 151)

---

> > > ### Public Comment · (anonymous) · 2018-10-30
> > > **This probably won't matter**
> > >
> > > I don't think it should matter what the Cleverhans defaults in the tutorials are ... as long as the authors correctly passed the lower and upper bounds, everything should work fine.

---

### Public Comment · (anonymous) · 2018-10-25
**Concerns about Table 4**

In table 4 you claim a L2 distortion of 187 is required to generate an adversarial example. Given that ImageNet images are 224x224x3 for ResNet, in the worst case converting a solid black image to a solid grey image would require a distortion of 193. Thus, a *mean* L2 distortion of 187 is exceptionally high. You might want to verify that you have properly implemented the attack back-propegating through the defense.

(Similarly to Table 5, L_infinity>0.5 is impossible by definition, and indicates the attack is failing.)

---

> ### Author Response · Authors · 2018-10-29
> **A note on normalization**
>
> Hi there, thank you for our interest in our work. In an image normalized to the range [0, 1], indeed an L2 distortion of 193 is the worst case. However, in section 4.1, we mention that our images are normalized to the range [-1, 1]. This is standard when working with Cleverhans to generate adversarial examples, as most pre-trained TensorFlow models accept data in the [-1, 1] range. With this range, the worst case perturbation has a magnitude of 387.9. Indicating that our defense is performing well.

---

> > ### Public Comment · (anonymous) · 2018-10-29
> > **You are claiming 100x what SOTA can achieve**
> >
> > Even in a range of [-1,1], the claim your paper is making is 100x larger than what state-of-the-art defenses can achieve. All prior defenses that have claimed even a 10x increase in distortion have been broken.
> >
> > Is it possible that your defense is actually two orders of magnitude stronger than what anyone else has been able to achieve? Yes.
> >
> > Is it likely? No.
> >
> > Please consider carefully evaluating the claims you have made. To make one concrete suggestion: you might want to try BPDA (Athalye et al. 2018) with your filter to see if it is breaking gradient descent. If possible, would you be willing to publish code for your pre-processing filter anonymously?

---

> > > ### Author Response · Authors · 2018-11-02
> > > **Comment on table 4**
> > >
> > > We didn't claim anywhere that we are 100x SOTA. Denoising approaches are known to do very well against L2 attacks. We also didn't claim this particular denoising approach (before adversarial training) is invulnerable. This is why we used adversarial training to improve this robustness as we mentioned in section 3.4.

---

> > > > ### Public Comment · (anonymous) · 2018-11-07
> > > > **[citation needed]**
> > > >
> > > > I am not aware of any other L2 defense on ImageNet that is robust to an L2 distortion greater than 10. Could you provide a citation for other denoising defenses?

---

### Meta-Review · Area_Chair1 · 2018-12-15
**Interesting proposal but the claims made are not well-supported**

**Confidence:** 5
**Recommendation:** Reject

**Metareview:**

The paper proposes a technique for defending against adversarial examples that relies on averaging pixels that are close to each other both in position and value. This approach seems to be an interesting preprocessing technique in the robust training pipeline. However, the actual claims made are not well-supported and, in fact, seem somewhat implausible.